analytical chemistry

sensor, carbon dots, abuse drugs, cathinone, cocaine

**Author for correspondence:**
Huan-Tsung Chang
e-mail: changht@ntu.edu.tw

This article has been edited by the Royal Society of Chemistry, including the commissioning, peer review process and editorial aspects up to the point of acceptance.

# Carbon dots functionalized papers for high-throughput sensing of 4-chloroethcathinone and its analogues in crime sites

Yao-Te Yen[1,2], Yu-Syuan Lin[1], Ting-Yueh Chen[2], San-Chong Chyueh[2] and Huan-Tsung Chang[2,3]

[1]Department of Chemistry, National Taiwan University, Taipei 10617, Taiwan, Republic of China
[2]Department of Forensic Science, Investigation Bureau, Ministry of Justice, Xindian Dist, New Taipei City 23149, Taiwan, Republic of China
[3]Department of Chemistry, Chung Yuan Christian University, Chungli District, Taoyuan City 32023, Taiwan, Republic of China

H-TC, 0000-0002-5393-1410

Sensitive and selective assays are demanded for quantitation of new psychoactive substances such as 4-chloroethcathinone that is a π-conjugated keto compound. Carbon dots (C-dots) prepared from L-arginine through a hydrothermal route have been used for quantitation of 4-chloroethcathinone in aqueous solution and on C-dot-functionalized papers (CDFPs). To prepare CDFPs, chromatography papers, each with a pattern of $8 \times 12$ circles (wells), are first fabricated through a solid-ink printing method and then the C-dots are coated into the wells. π-Conjugated keto or ester compounds induce photoluminescence quenching of C-dots through an electron transfer process. At pH 7.0, the CDFPs allow screening of abused drugs such as cocaine, heroin and cathinones. Because of poor solubility of heroin and cocaine at pH 11.0, the C-dot probe is selective for cathinones. The C-dots in aqueous solution and CDFPs at pH 11.0 allow quantitation of 4-chloroethcathinone down to 1.73 mM and 0.14 mM, respectively. Our sensing system consisting of a portable UV-lamp, a smartphone and a low-cost CDFP has been used to detect cathinones, cocaine and heroin at pH 7.0, showing its potential for screening of these drugs in crime sites.

## 1. Introduction

Carbon dots (C-dots) have become interesting sensing materials in recent years due to their ease in preparation, stability and strong

photoluminescence (PL) [1–3]. Although C-dots are usually not as bright as semiconductor quantum dots, they provide many advantages, including low toxicity, biocompatibility, as well as stability against salt and photoirradiation [4]. They have been used for sensing of analytes such as $Cr^{4+}$, $Co^{2+}$, $Hg^{2+}$, $NO_2^-$, vitamin $B_{12}$, trinitrophenol, dopamine and iodine species [5–12]. Owing to the existence of many hydrophilic functional groups, including hydroxyl, amino and carboxylate, on their surfaces, C-dots are highly dispersible in aqueous solution and can be functionalized with recognition elements, making them suitable for sensing [13–18].

Although C-dots have become important sensing materials, their use in quantitation of abused drugs has not been reported. Cocaine, heroin, ketamine, marijuana and methamphetamine are common abused drugs that have caused serious social problems all around the world [19]. A few commercial immunoassays are available for detection of common abused drugs in criminal sites [20–23]. Most of them are specific and sensitive, but high cost, short shelf-life, and temperature or moisture interference are problems. Moreover, sensitive and selective assays for sensing of new psychoactive substances like cathinones are usually unavailable, which has put great pressure on law enforcement. Cathinones are π-conjugated keto compounds that have grown steadily as abused drugs around the world. Cathinone abusing has a similar symptom to that of methamphetamine, but it causes more addictive and mental-reliable problems [24]. Thus, developing low-cost nanomaterials-based sensitive sensors for cathinones is important.

Herein, we report a C-dots-based turn-off PL sensing assay for the detection of cathinones like 4-chloroethcathinone. The C-dots were prepared from L-arginine through a hydrothermal route [25–28]. We investigated pH effects on the sensitivity and selectivity for quantitation of 4-chloroethcathinone in aqueous solution. The sensing mechanism of C-dots for 4-chloroethcathinone was proposed based on the absorption, PL and cyclic voltammetry (CV) data. The C-dots were further used to fabricate C-dot-functionalized papers (CDFPs) for screening of cathinones. Our results show great potential of the CDFPs for detection of cathinones in crime sites.

# 2. Material and methods

## 2.1. Materials

Acetone, L-arginine, ferrocene, formaldehyde, fructose, γ-butyrolactone, glucose, 1-methyl-2-pyrrolidone, quinine sulfate monohydrate, sodium borohydride, sucrose and tetrabutyl ammonium hexafluorophosphate were purchased from Sigma Aldrich (Milwaukee, WI, USA). Monobasic and dibasic sodium phosphates obtained from J. T. Baker (Phillipsburg, NJ, USA) were used to prepare sodium phosphate buffers (100 mM, pH 7.0 and pH 11.0). A Milli-Q ultrapure water system (Millipore, Billerica, MA, USA) was used to produce ultrapure water (18.2 mΩ cm) for preparation of all aqueous solutions used in this study. Butylone, 4-chloroethcathinone, 4-chloromethcathinone, cocaine, ephedrine, ethylone, heroin, ketamine, methamphetamine and mexedrone are all in hydrochloride salt forms, which were from the Ministry of Justice Investigation Bureau, Taiwan. Their high purities were confirmed by GC–MS and NMR measurements. Chromatography paper (20 × 20 cm) was purchased from Whatman (Little Chalfont, Buckinghamshire, UK).

## 2.2. Synthesis of C-dots

C-dots were synthesized from L-arginine aqueous solution (0.3 M, 20 ml) in a Teflon-lined stainless-steel container through a hydrothermal route [25–28]. After heating at 240°C for 14 h, the resulting brownish-yellow solution was cooled to ambient temperature (25°C) and then filtered through a 0.2 mm nylon membrane to remove large particles. Subsequently, the resulting solution was purified against ultrapure water through a dialysis membrane (MWCO 3.5 kD) from Spectra Labs (Rancho Dominguez, CA, USA) for 24 h to eliminate the fluorescence impurities [29,30]. To determine the amount of C-dot per ml of aqueous solution, 1 ml of C-dot solution was dried and weighted. The concentration of C-dots in the as-prepared solution is *ca* 13.2 mg ml$^{-1}$. The as-prepared C-dot solution was stored at 4°C in the dark prior to use. The quantum yield ($\Phi_f$) of C-dots dispersed in ultrapure water was calculated by comparing their integrated PL intensity (excited at 360 nm) and absorbance at 360 nm with those of quinine sulfate ($\Phi_f = 0.54$) that was dissolved in 0.1 M $H_2SO_4$. The absorbance values of the two solutions in 1 cm (optical path length) cuvettes were kept under 0.1 at their corresponding maximum excitation wavelengths to minimize the re-absorption effects.

## 2.3. Characterization of C-dots

Transmission electron microscopy (TEM) images of C-dots were recorded using FEI Tecnai-G2-F20 from GCE Market (Blackwood, NJ, USA) operated at 200 kV. High-resolution TEM (HRTEM) images of C-dots were measured using FEI Talos F200× from Thermo Fisher Scientific (Waltham, MA, USA). Before conducting TEM measurements, the purified C-dots were diluted 10-fold with ultrapure water. The diluted C-dots were deposited on 400-mesh carbon-coated Cu grids and excess solvent was evaporated at ambient temperature and pressure. The surface elements of C-dots were investigated through X-ray photoelectron spectroscopy (XPS) from JEOL (JPS-9030, Tokyo, Japan) with Al K$\alpha$ X-ray radiation. A micro-Raman spectrometer (UniRam Series, ProTrusTech, Tainan, Taiwan) equipped with a 50× objective (numerical aperture 0.35) and a laser emitting at 532 nm was used to collect their Raman spectra. For each measurement, the data acquisition time was 2 s for 10 accumulations. The functional groups on the surface of C-dots were analysed based on their Fourier transform infrared (FTIR) spectra using a Thermo Scientific spectrometer (Nicolet iS5, Waltham, MA, USA. A UV–Vis spectrophotometer (Evolution 220) from Thermo Fisher Scientific (Waltham, MA, USA) was used to record the absorption spectra of C-dot solutions. The PL spectra of the as-prepared C-dots were recorded using a Cary Eclipse PL spectrophotometer from Varian (Palo Alto, CA, USA) that was excited at the wavelengths from 250 to 440 nm.

## 2.4. Detection of 4-chloroethcathinone and its analogues in aqueous solution

Sodium phosphate solutions (final concentration 90 mM, pH 11.0) were used to evaluate the sensitivity of C-dot probe for detection of 4-chloroethcathinone. For example, aliquots (40 µl) of the purified C-dot solution were added into sodium phosphate buffers (360 µl, 100 mM, pH 11.0) containing various concentrations of 4-chloroethcathinone. The PL spectra were recorded after the reaction proceeded for 1 min. To investigate the analyte induced PL quenching in the presence of a reducing agent, sodium borohydride (15.1 mg) was added into phosphate solution (4 ml, 100 mM, pH 11.0) containing 27.8 mM of 4-chloroethcathinone. After the reaction of 4-chloroethcathinone with sodium borohydride for 20 min, the purified C-dot solution (40 µl) was added into the mixture (360 µl). Its PL spectrum was recorded after the reaction for 1 min. Selectivity test was carried out in sodium phosphate solutions (90 mM, pH 7.0 or pH 11.0) containing other common abused drugs, including cocaine, heroin, methamphetamine, ketamine and ephedrine. In addition, additives, including glucose, sucrose and fructose were also tested for their interference. Moreover, cathinone analogues, including 4-chloromethcathinone, butylone, ethylone and mexedrone, were also used for detection. To understand the analyte-induced PL quenching, two π-conjugated ketones—γ-butyrolactone and 1-methyl-2-pyrrolidone—were tested separately. The final concentrations of the tested drugs are 25 mM in sodium phosphate buffers (90 mM, pH 7.0 or 11.0), except that for cocaine, heroin and ketamine in sodium phosphate buffers (90 mM, pH 11.0) are saturated, which are 15.1, 12.8 and 11.6 mM, respectively. All experiments were performed under ambient conditions (25°C, 1 atm).

## 2.5. Sensing mechanism

To propose a sensing mechanism of C-dots for the detection of the tested abused drugs, time-resolved absorption and emission spectroscopy (NS010, Pascher Instrument, Sweden) was used to measure the PL lifetime of C-dots (1.32 mg ml$^{-1}$) in sodium phosphate buffers (90 mM, pH 11.0) without/with containing 4-chloroethcathinone (15 mM). In addition, the highest occupied molecular orbitals (HOMOs) and lowest unoccupied molecular orbitals (LUMOs) for C-dots, 4-chloroethcathinone, cocaine and ephedrine were determined based on their absorption and CV data. Tetrabutyl ammonium hexafluorophosphate acetonitrile solution (0.1 M, 10 ml) was used separately to prepare solutions of C-dots (0.66 mg ml$^{-1}$), 4-chloroethcathinone (10 mM), cocaine (10 mM), ephedrine (1 mM) and ferrocene (1 mM). They were subjected to CV measurements using a CHI 760D electrochemical workstation from CH instruments (Austin, TX, USA) in a positive mode with a sweeping rate of 0.1 V s$^{-1}$ under the ambient condition. Glassy carbon electrode was used as a working electrode, with an Ag/AgCl (saturated by KCl) electrode as a reference electrode and a platinum wire as a counter electrode. Absorption spectra of 4-chloroethcathinone, cocaine and ephedrine prepared in sodium phosphate buffer solution (100 mM, pH 7.0), each at the concentration of 0.1 mM, were recorded to estimate their energy gaps.

## 2.6. Fabrication of C-dot-functionalized papers

CDFPs were prepared by depositing C-dots onto chromatography papers that had been patterned using a solid-ink printing method [31,32]. A pattern (12 × 8 circles) was printed onto one side of the chromatography paper (20 × 20 cm) using a wax printer (Xerox 8570, Fuji, Japan). The diameter of each circle (well) is 5 mm. Subsequently, the chromatography paper was flipped and wax was printed onto the back side. The wax-patterned chromatography paper was placed on a hot plate and heated at 140°C for 1 min to allow the wax to penetrate the paper to form hydrophobic barriers. Aliquot (4 µl) of the purified C-dot solution was placed onto each well in the as-prepared chromatography paper. Excess solvent was evaporated at 50°C in an oven.

## 2.7. Detection of 4-chloroethcathinone on CDFPs

Aliquots (10 µl each well) of various concentrations of 4-chloroethcathinone dissolved in sodium phosphate buffer solution (100 mM, pH 11.0) were dropped separately onto the wells in the CDFPs. After the excess solvent was evaporated, the PL spectrum of the C-dots in each well and the PL image of the CDFPs were recorded at an excitation wavelength of 254 nm using a microplate fluorometer (Synergy 4 Multi-Mode Microplate Reader) from BioTek instruments (Winooski, VT, USA). A smartphone (HTC, pro 10, Taiwan) was used to take the PL images when the CDFPs were excited at 254 nm using a portable UV-lamp.

Selectivity test was performed by adding heroin, cocaine, ketamine, methamphetamine or ephedrine (10 µl, 25.0 mM or saturated) prepared in sodium phosphate solution (100 mM, pH 11.0 or 7.0) onto each well in the CDFPs. After 10 min, the PL spectrum of the C-dots in each well and the PL image of CDFP were recorded when excited at 254 nm. To test possible interference from additives, 4-chloroethcathinone mixed with glucose at various concentrations (w/w% from 50 to 95%) were prepared. The mixture (20 mg) was dissolved in 500 µl of sodium phosphate solution (100 mM, pH 11.0). After being mixed for 1 min, aliquots (10 µl) of the mixtures were added onto the wells of the CDFPs. Their PL spectra and images were recorded separately after the excess solvent was evaporated. Sensing of 4-chloroethcathinone in urine was tested to validate practicality of the assay. Aliquots of 4-chloroethcathinone were spiked to blank urines, with different final concentrations ranging from 500 to 12 500 ng ml$^{-1}$. Each of the 50 ml spiked urine samples was transferred to a SPE column (Strata-X-Drug B, Phenomenex, USA) that had been previously equilibrated with 4 ml of methanol and then with 4 ml of sodium phosphate solution (100 mM, pH 9.0). Subsequently, the column was washed with 10 ml of H$_2$O, 10 ml of sodium phosphate solution (100 mM, pH 4.0) and 10 ml of methanol. Finally, the adsorbed solutes were eluted using 3 ml of dichloromethanol/isopropyl alcohol/NH$_4$OH (80 : 15 : 5; v/v). After evaporating to dryness under a nitrogen stream at 40°C, the residue was dissolved in 100 µl of sodium phosphate solution (100 mM, pH 11.0). Aliquots (10 µl) of the mixtures were added onto wells of the CDFPs, and their PL spectra were recorded separately after the excess solvent was evaporated.

# 3. Results and discussion

## 3.1. Properties of C-dots

According to the literature [4], the highly water-dispersible C-dots prepared from L-arginine were suggested to be formed through the oxidation, carbonization, polymerization and passivation processes. The TEM image of as-prepared C-dots displayed in figure 1a shows that they are uniform spheres with a mean diameter of 4.4 nm (100 counts). The HRTEM image of C-dots displayed in electronic supplementary material, figure S1 shows the lattice spacing of 0.38 nm, which is corresponding to the (002) plane of graphitic carbon, revealing the formation of highly crystallized C-dots [33]. The XPS spectrum of C-dots displayed in figure 1b displays three identified peaks at 285, 400 and 532 eV, corresponding to C1s, N1s and O1s, respectively. Furthermore, four deconvoluted C1s peaks were obtained at 284.8, 285.4, 286.3 and 288.2 eV, which are attributed to C–C, C–N, C–O and C=O, respectively (electronic supplementary material, figure S2A). The N1s peak was deconvoluted to three peaks at 399.6, 400.4 and 401.9 eV, corresponding to C–N–C, N–(C)$_3$ and N–H bonds, respectively (electronic supplementary material, figure S2B). The O1s spectrum displays two major peaks at 531.5 and 532.7 eV, corresponding to C–O and C–OH/C–O–C, respectively (electronic

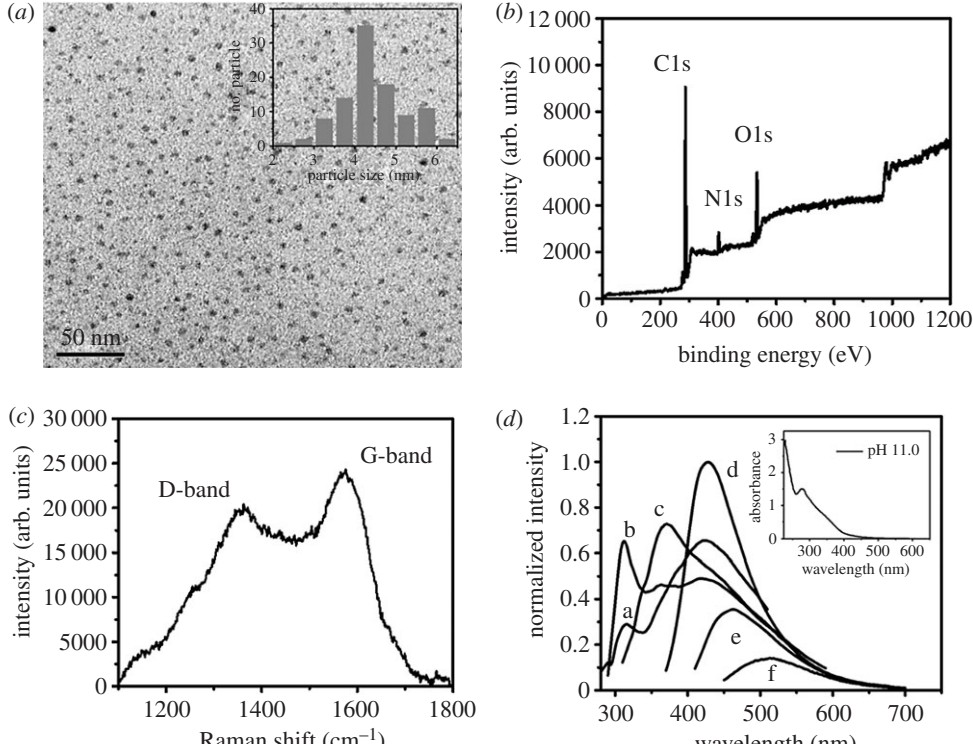

**Figure 1.** (a) TEM image, (b) XPS spectrum and (c) Raman spectrum of C-dots. (d) Emission spectrum of C-dots (0.53 mg ml$^{-1}$) in sodium phosphate buffer (90 mM, pH 11.0) when excited at various wavelengths. Excitation wavelengths for a, b, c, d, e and f are 260, 280, 300, 360 400 and 420 nm, respectively. The Inset in (a): the size distribution of particles with a diameter range of 2–7 nm (100 counts). The Inset in (d): UV–Vis absorption spectrum of the corresponding C-dot solution.

supplementary material, figure S2C). The Raman spectrum displayed in figure 1c shows the D-band at 1358 cm$^{-1}$ for the vibrations of carbon atoms with dangling bonds in the termination plane of disordered graphite or glassy carbon, as well as the G-band at 1572 cm$^{-1}$ for the in-plane stretching of sp$^2$ carbon in the rings. Their intensity ratio ($I_D/I_G$) is 0.8, which is similar to the most reported C-dots, supporting C-dots containing sp$^2$ (core) and sp$^3$ (surface) hybridized carbons [34–37]. Like most reported C-dots [28,38–41], the as-prepared C-dots exhibit interesting excitation-wavelength-dependence emission properties when excited at wavelengths ranging from 300 to 420 nm, as shown in figure 1d [42]. The strongest PL intensity was obtained at 430 nm when excited at 360 nm, with a quantum yield of about 16.0%. When separately excited at 260 and 280 nm, the PL bands centre at 316 and 312 nm, respectively, which are almost excitation independent. The two PL bands are single emissive translations, which are mainly due to the core of the C-dots [42]. The absorption spectrum (inset to figure 1d) of the C-dots exhibits an absorption band at the wavelengths around 240 nm and a shoulder at around 330 nm, corresponding to π–π* and n–π* transitions, respectively [40,43,44]. The FTIR spectrum of the C-dots displayed in electronic supplementary material, figure S3 exhibits a C=O vibrational stretching band at 1651 cm$^{-1}$, a C–H stretching band at 2952 cm$^{-1}$ and a strong O–H vibrational stretching band at 3362 cm$^{-1}$, which is similar to arginine (precursor).

## 3.2. Quantitation of 4-chloroethcathinone

The PL intensities of C-dot solutions at 430 nm over the pH range from 5.0 to 11.0 are similar (less than 10% variations) when excited at 365 nm. Like most reported C-dots [45], they are quite stable against salt (up to 1.0 M NaCl) and photoirradiation (less than 20% PL change under irradiation from a light with intensity of 3 W at 365 nm for 2 h). Over pH values from 5.0 to 11.0, 4-chloroethcathinone (25.0 mM) induced PL quenching of C-dots increases upon increasing pH value, mainly because of increases in its oxidation strength [46,47]. Figure 2a displays that 4-chloroethcathinone-induced PL quenching of C-dots at pH 11.0 increases upon increasing its concentration up to 25.0 mM. As shown in the inset to figure 2a, the relationship of $(F_0 - F)/F_0$ of C-dots is linear ($Y = 0.028X + 0.023$, $R^2 = 0.98$) with the

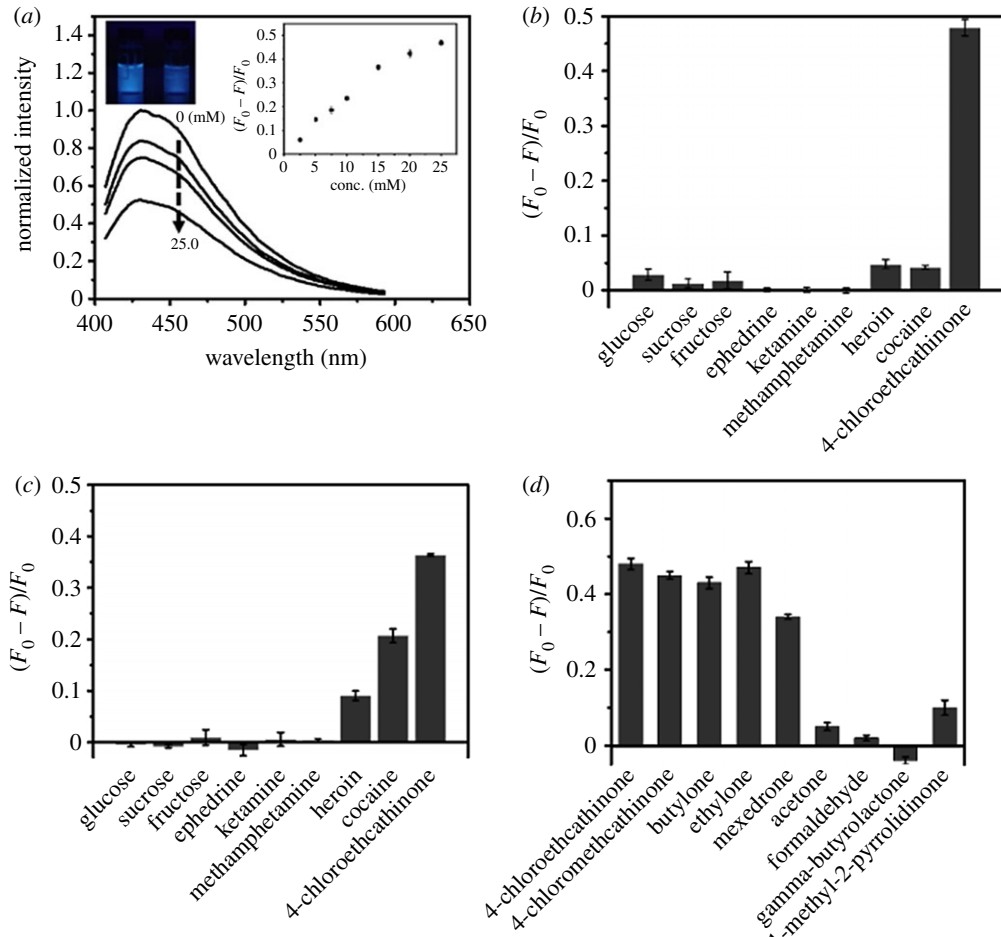

**Figure 2.** Detection of 4-chloroethcathinone using C-dots. (*a*) Sensitivity for 4-chloroethcathinone at pH 11.0, and (*b,c*) selectivity towards 4-chloroethcathinone over other common illicit drugs and additives at pH values of (*b*) 11.0 and (*c*) 7.0. Sodium phosphate buffers (90 mM, pH 11.0 or 7.0) were used to prepare the probe solutions. The concentrations of glucose, sucrose, fructose, ephedrine, methamphetamine and 4-chloroethcathinone are all 25.0 mM, and the others are saturated. (*d*) Selectivity towards π-conjugated keto compounds such as 4-chloroethcathinone and other cathinone-drugs over other tested chemicals containing acetone, formaldehyde, γ-butyrolactone and 1-methyl-2-pyrrolidinone at pH values of 11.0. The concentrations of analytes are all 25.0 mM. Error bars represent standard deviations from three repeated experiments.

concentration of 4-chloroethcathinone over the range from 2.5 to 15.0 mM, in which $F$ and $F_0$ are the PL intensities of the C-dots in the presence and absence of 4-chloroethcathinone, respectively. A detection limit of 1.73 mM (0.43 mg ml$^{-1}$) for 4-chloroethcathinone was obtained according to 3$\delta$/slope, in which $\delta$ and slope represent the deviation of $(F_0 - F)/F_0$ for blank test and the slope of the calibration curve, respectively. Figure 2*b* shows that 4-chloroethcathinone-induced PL quenching at pH 11.0 is higher than that at pH 7.0. More importantly, some common abused drugs such as cocaine, ephedrine, heroin, ketamine and methamphetamine, as well as some popular additives such as glucose, fructose and sucrose did not cause significant PL quenching at pH 11.0. Cocaine and heroin do induce about 21% and 9% decreases in the PL intensity of C-dots at pH 7.0 as shown in figure 2*c*. Because they are slightly soluble at pH 11.0, they do not induce significant PL quenching of C-dots. Based on the fact that 4-chloroethcathinone contains a π-conjugated keto group, while heroin and cocaine both contain a π-conjugated ester group, as shown in electronic supplementary material, figure S4, we suggested that π-conjugated keto or ester group plays an important role in PL quenching of C-dots. Since 4-chloroethcathinone relative to heroin and cocaine induces larger PL quenching, π-conjugated keto group interacts more strongly with the C-dots. To support our suggestion, other π-conjugated keto compounds, such as 4-chloromethcathinone, ethylone, butylone and mexedrone, and compounds containing keto, aldehyde, ester or amide group, such as acetone, formaldehyde, γ-butyrolactone and 1-methyl-2-pyrrolidinone, were tested at pH 11.0. As shown in

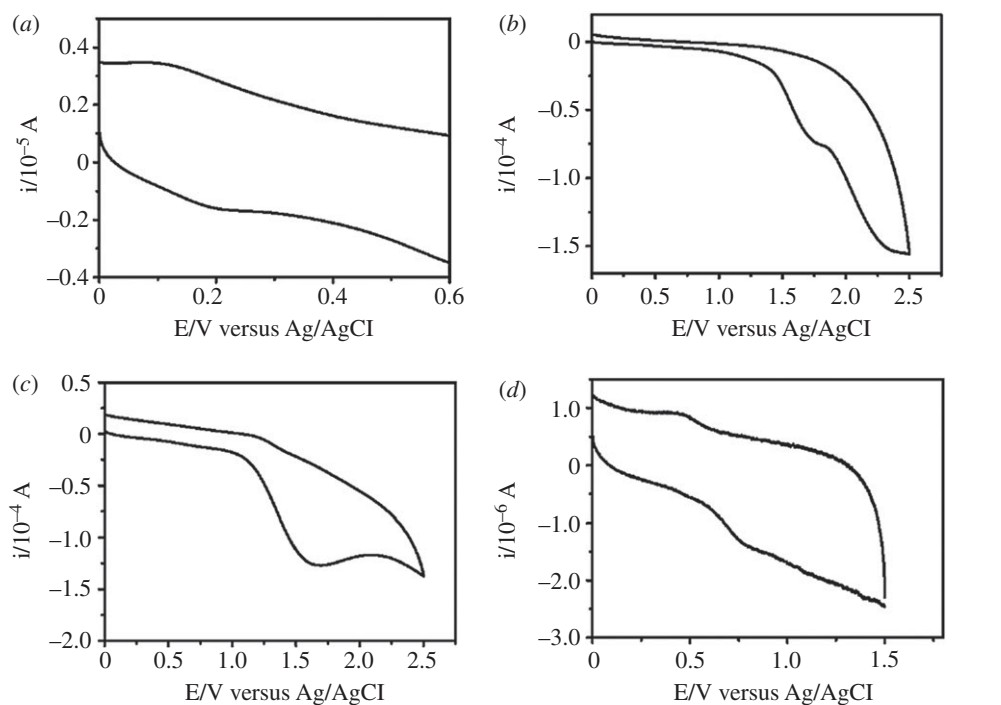

**Figure 3.** Cyclic voltammograms of (a) 0.66 mg ml$^{-1}$ of C-dots, (b) 10 mM of 4-chloroethcathinone, (c) 10 mM of cocaine and (d) 1 mM of ephedrine in ACN. Cyclic voltammetry was conducted in a positive mode with a sweeping rate of 0.1 V s$^{-1}$ at ambient condition.

figure 2d, only π-conjugated keto compounds induced significant PL quenching. The percentages of decreases in PL intensity of C-dots induced by 4-chloroethcathinone, 4-chloromethcathinone, ethylone, butylone and mexedrone are 48%, 45%, 43%, 47% and 34%, respectively. The present approach over the reported methods provides advantages of simplicity and low cost for quantitation of cathinone-drugs [48,49]. These compounds are all good electron acceptors and they thus were reduced from a keto form to a hydroxyl form by C-dots [50,51]. The results support that the C-dot probe is selective to π-conjugated keto compounds at pH 11.0. In addition, metal ion species such as $Fe^{3+}$, $Co^{2+}$, $Cr_2O_7^{2-}$, $Cd^{2+}$, $Hg^{2+}$, $Ag^+$ and $Au^{3+}$ are not present in seized street samples, because of these metal ions are not used as catalysts during syntheses or production of these drugs.

## 3.3. Sensing mechanism

The PL quenching of C-dots induced by 4-chloroethcathinone follows the Stern–Volmer equation, as shown in electronic supplementary material, figure S5, revealing a dynamic quenching mechanism. In other words, either resonance energy transfer or electron transfer occurred between 4-chloroethcathinone and C-dots. From the slope of the linear plot, $K_{SV}$ was determined to be 0.0379. Since 4-chloroethcathinone has no absorption above the wavelength of 300 nm, its absorption spectrum does not overlap with the emission spectrum of C-dots, ruling out a resonance energy transfer mechanism. To investigate an electron transfer mechanism occurring between 4-chloroethcathinone and the C-dots, the lifetimes of C-dots without/with containing 4-chloroethcathine (15 mM) at emission/excitation wavelength of 430/360 nm were determined to be 13 and 11 ns (electronic supplementary material, figure S6), respectively [52–54]. In addition, CV and UV–vis absorption were conducted to determine the HOMO and LUMO levels of the C-dots and 4-chloroethcathinone [55–57]. As controls, the HOMO and LUMO levels of cocaine (induces small PL quenching) and ephedrine (does not induce PL quenching) were also determined. From the CV data (figure 3), the oxidation potentials of C-dots, 4-chloroethcathinone, cocaine and ephedrine were determined to be 0.19, 1.69, 1.61 and 0.75 eV, respectively. Their HOMO levels were calculated using the equation of $E_{HOMO} = -(4.8 - E_{OX,Fc/Fc+} + E_{OX})$, where $E_{OX,Fc/Fc+}$ is the oxidation potential of ferrocene/ferrocene$^+$ and $E_{OX}$ is the oxidation potential of the tested chemicals. On the other hand, their LUMO levels were calculated using the equation of $E_{LUMO} = E_{HOMO} + E_{0-0}^{onset}$, where $E_{0-0}^{onset}$

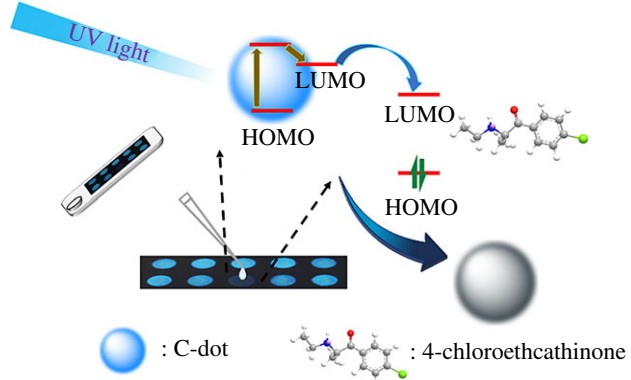

**Scheme 1.** Schematic illustration of C-dot-functionalized paper (CDFP) for detection of 4-chloroethcathinone.

**Table 1.** HOMO, LUMO and band gap of C-dots and analytes.

| sample | HOMO (eV) | LUMO (eV) | band gap (eV) |
|---|---|---|---|
| C-dots | −4.51 | −1.48 | 3.03 |
| cocaine | −5.93 | −1.80 | 4.13 |
| 4-chloroethcathinone | −6.01 | −1.95 | 4.06 |
| ephedrine | −5.07 | −0.61 | 4.46 |

represents the 0–0 energy, which is defined as the lowest energy transition estimated by the longest absorption wavelength ($\lambda$onset) of the tested chemicals (electronic supplementary material, figure S7) [58,59]. Table 1 summarizes the values of their HOMO, LUMO and energy gap, revealing that the LUMO levels of 4-chloroethcathinone and cocaine are lower than that of C-dots. Thus, electron transfer from the C-dots to the two chemicals could occur, supporting the PL quenching through electron transfer [60–63]. Moreover, to further support the electron transfer mechanism, 4-chloroethcathinone (27.8 mM) reacted with sodium borohydride (0.1 M) prior to its addition to the C-dot solution at pH 11.0. Sodium borohydride reduced the keto group of 4-chloroethcathinone to a hydroxyl group [51], and thus the mixture only induced a slight decrease (9%) in the PL intensity of the C-dots. It is noted that sodium borohydride itself caused about 12% decrease in the PL intensity of C-dots, mainly because it also induced reduction of the surface residues like keto or aldehyde groups of C-dots through nucleophilic attack by the hydride anion.

Based on the absorption, PL and CV data, the HOMO (−4.51 eV), LUMO (−1.48 eV) and energy gap (3.03 eV) of C-dots were obtained, which are close to that (HOMO: −4.81 eV, LUMO: −1.74 eV and energy gap: 3.08 eV) from DFT calculation reported in the literature [64]. Accordingly, a sensing mechanism of C-dots for 4-chloroethcathinone is proposed as displayed in Scheme 1. The energies of light corresponding to the wavelengths at 260 and 360 nm are 4.77 and 3.44 eV, which are higher than the energy gap between LUMO and HOMO of C-dots. Thus, upon excitation at the two wavelengths, C-dot is excited to higher energy levels and the excited electron then transfers to its LUMO through a non-emission relaxing process. The excited electron (energy) in the LUMO of the C dots further transfers to the LUMO of 4-chloroethcathinone, leading to PL quenching of C-dots.

## 3.4. Detection of 4-chloroethcathinone using CDFPs

Figure 4a shows quantitation of 4-chloroethcathinone at the concentrations ranging from 0.5 to 25.0 mM using CDFPs when excited at 254 nm with a portable UV-lamp. Although the PL of C-dot is stronger when excited at 360 nm, 254 nm was selected, mainly because of lower emission background from the paper. Like in the solution phase, the PL intensity of C-dots on the CDFP decreases upon increasing the concentration of 4-chloroethcathinone. A linear range ($R^2 = 0.95$) was obtained over the concentration range from 0.5 to 10.0 mM, with a detection limit of 0.14 mM (0.03 mg ml$^{-1}$) for

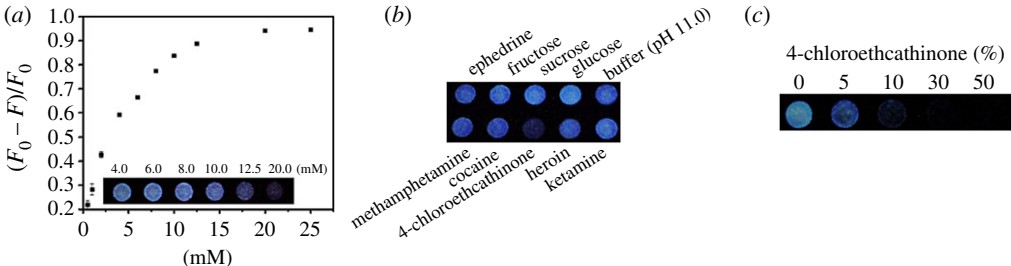

**Figure 4.** CDFPs for detection of 4-chloroethcathinone. At pH 11.0 using a portable UV-lamp (254 nm) as the light source. (*a*) Sensitivity, (*b*) selectivity and (*c*) glucose tolerance. Inset to (*a*) PL images of CDFPs in the presence of various concentration of 4-chloroethcathinone. (*b*) The concentrations of the tested chemical are 25 mM, besides that are saturated for cocaine, heroin and ketamine. (*c*) w/w% of 4-chloroethcathinone in the presence of glucose. A smartphone was used to record the PL images.

4-chloroethcathinone. Inset to figure 4*a* is the PL image of C-dots on the CDFP in the presence of various concentrations of 4-chloroethcathinone, which was taken using a smart phone. The detection was completed within 5 min, showing its advantages of high throughput and its great potential for on-site quantitation of 4-chloroethcathinone. When compared to the assay conducted in the solution phase, CDFPs provide a lower detection limit, mainly because the analyte and C-dots were concentrated in each well. In addition to having better sensitivity, it is more convenient for the law enforcement officers to identify whether illegal drugs were used in crime sites when using the CDFPs.

Abused drugs are often mixed with glucose, and thus it is important to test the effect of glucose on the quantitation of 4-chloroethcathinone. Figure 4*c* displays that 4-chloroethcathinone induced significant PL quenching of C-dot in the presence of glucose at the w/w% up to 95%. At the w/w% of 4-chloroethcathinone higher than 10%, the PL of C-dots is almost quenched completely. It is important to note that using a smart phone, as low as 5 w/w% of 4-chloroethcathinone can be detected. It is rare that an abused drug is mixed with such a high concentration of glucose. As a control, glucose at the concentration up to 100% did not induce PL quenching. Because π-conjugated keto compounds induced PL quenching can be observed clearly by the naked eye using a portable UV-lamp, the CDFP holds great potential for on-site detection of cathinones. In addition, sensing of 4-chloroethcathinone in urine was tested using CDFPs. A linear range ($R^2 = 0.98$) was obtained over the concentration range from 2000 to 12 500 ng ml$^{-1}$ with a detection limit of 1300 ng ml$^{-1}$, which shows the potential of this low-cost assay for sensing of cathinones in urine samples.

## 4. Conclusion

C-dots prepared from L-arginine are selective for quantitation of π-conjugated keto compounds at pH 11.0. At pH 7.0, the C-dots are selective for abused drugs containing a π-conjugated keto or ester group. The sensing mechanism is based on analyte-induced PL quenching through an electron transfer process. Our results show that the C-dot probe can be used for screening of 4-chloroethcathinone, its analogues, and common drugs such as cocaine and heroin at pH 7.0, and for screening of cathinones at pH 11.0. Furthermore, CDFPs were successfully applied for selective quantitation of 4-chloroethcathinone and its analogues using a low-cost smartphone and a portable UV-lamp. This simple, low-cost and selective CDFP allows law enforcement officers to rapidly screen cathinones in crime sites.

Ethics. There are no conflicts to declare.
Data accessibility. Electronic supplementary material can be found online on the website of the publisher.
Authors' contributions. Y.-S.L. helped to study and analyse the data of carbon dot for stability. T.-Y.C. helped to revise the manuscript. S.-C.C. and H.-T.C. revised the manuscript and coordinated the study.
Competing interests. We declare we have no competing interests
Funding. We are grateful to the Ministry of Science and Technology (MOST) and the Ministry of Justice (MOJ) of Taiwan for providing financial support for this study under contracts NSC 107-2622-M-002-002-CC2 and 108-1301-05-17-05.
Acknowledgements. The assistance of Ms Y.-Y.Y. and Ms C.-Y.L. from the Instrument Center of National Taiwan University (NTU) for TEM measurement is appreciated.

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
