## [Reviewer comments · Royal Society Open Science]

Review History

RSOS-191017.R0 (Original submission)

Review form: Reviewer 1

Is the manuscript scientifically sound in its present form?

Yes

Are the interpretations and conclusions justified by the results?

No

Is the language acceptable?

Yes

Is it clear how to access all supporting data?

No

Do you have any ethical concerns with this paper?

No

Have you any concerns about statistical analyses in this paper?

No

Recommendation?

Major revision is needed (please make suggestions in comments)

Comments to the Author(s)

The manuscript reports carbon dots functionalized papers for high-throughput sensing of 4-Chloroethcathinone and its analogues in crime sites. There are some interesting results. And I think there is a wide reader region to have interests in this kind of material. I recommend accepting this work. I wish the authors could give some revision before it could be accepted.

- 1) Please explain the meaning of The Raman spectrum. The description of The Raman spectrum in the article is not detailed enough.
- 2) Characterization of CDs lacks FTIR and XPS data
- 3) The author referred the relationship of logarithm of $(F_0-F)/F_0$ of C-dots is linear with the concentration of 4-chloroethcathinone over the range from 2.5 to 15.0 Mm, but we can't get this relationship from the Fig.1A.
- 4) For "To support our suggestion, other.....they thus were reduced from a keto form to a hydroxyl form by C-dots.", the author should give the picture of the data "48%, 45%, 43%, 47%, and 34%", and the comparison of the published article to state the advantages.
- 5) For "To investigate an electron transfer mechanism occurring between 4-chloroethcathinone and the C-dots, the lifetimes of C-dots without with containing 4-chloroethcathine (15 mM) at emission excitation wavelength of 430/360 nm were determined to be 13 and 11 ns, respectively.", the author should give the picture of the data "13 and 11 ns".
- 6) The author referred Based on the absorption, PL, and CV data, a sensing mechanism of C-dots for 4-chloroethcathinone is proposed as displayed in Scheme 1, but we can find the absorption data
- 7) The authors write "Moreover, to further support the electron transfer mechanism.....It is noted that sodium borohydride itself caused about 12% decrease in the PL intensity of C-dots." Please explain the mechanism of 4-chloroethcathinone reacted with sodium borohydride.

Review form: Reviewer 2**Is the manuscript scientifically sound in its present form?**

No

Are the interpretations and conclusions justified by the results?

Yes

Is the language acceptable?

Yes

Is it clear how to access all supporting data?

Yes

Do you have any ethical concerns with this paper?

No

Have you any concerns about statistical analyses in this paper?

Yes

Recommendation?

Major revision is needed (please make suggestions in comments)

Comments to the Author(s)

In this manuscript, Yen et al reported the preparation of a novel kind of C-dot functionalized paper that can be used for detecting 4-Chloroethcathinone and its analogues with a high selectivity. The topic is interesting and the results sounds good. Unfortunately, I cannot accept this paper for publication at its present form, because the manuscript was poorly prepared, besides this, I also have some serious concerns about the data analysis as follows. In my opinion, a major revision should be done before it can be re-considered for publication in the Royal Society Open Science journal.

(1) On page 5, the authors claimed that “The TEM image of as-prepared C-dots displayed in Fig. S1 (Supporting Information) shows that they are uniform and monodispersed spheres with a mean diameter of 5 nm (100 counts)”. However, the authors did not provide the size distribution analysis curve on the basis of the TEM image shown in Fig. S1;

(2) Fig. S2 (Supporting Information) shows “the D-band at 1320 cm^{-1} and the G-band at 1590 cm^{-1} ”, however, there is an obvious Raman signal at 1450 cm^{-1} between D and G bands, what does this represent for? The authors need to give an assignment of this Raman signal.

(3) Still on page 5, “Their intensity ratio (ID/IG) is 1.4, which is similar to the most reported C-dots, supporting C-dots containing sp² (core) and sp³ (surface) hybridized carbons.[33-36]” I think the authors have missed many related papers in explanation of ID/IG ratio of carbon dots, because most of the carbon dots with sp²-hybridized carbon cores show the ID/IG ratios in the range of 0.3 ~ 0.8. And the listed references are not adequate here.

(4) On page 3, C-dots were synthesized from L-arginine aqueous solution in a Teflon-lined stainless-steel container through a hydrothermal route after heating at 240 °C for 14 h. It is reasonable to expect that the as-synthesized C-dots are highly crystallized at so high a temperature, however, The authors did not provide a high-resolution TEM image of the carbon core, and Raman spectrum of the C-dots as shown in Fig. S2 is very different from that of the purified carbon dots as previously reported. It suggests the presence of a sort of impurity inside the C-dots.

(5) On page 5, the authors thought D-band at 1320 cm^{-1} representing for the edge defects, but this is not correct. Since the edge defects usually exist in graphene quantum dots, herein no any evidence can demonstrate that the obtained C-dots are graphene quantum dots.

(6) Although using CV and UV-vis absorption methods to determine the HOMO and LUMO levels of the C-dots (Fig. 2A) is very interesting, according to Table 1, the band gap was evaluated to be 3.03 eV, this value should be compared to those obtained from theoretical calculation in the literature.

Decision letter (RSOS-191017.R0)

01-Jul-2019

Dear Dr Chang:

Title: Carbon Dots Functionalized Papers for High-Throughput Sensing of 4-Chloroethcathinone and its analogues in Crime Sites

Manuscript ID: RSOS-191017

The editor assigned to your manuscript has now received comments from reviewers. We would like you to revise your paper in accordance with the referee and Subject Editor suggestions which can be found below (not including confidential reports to the Editor). Please note this decision does not guarantee eventual acceptance.

Please submit your revised paper before 24-Jul-2019. Please note that the revision deadline will expire at 00.00am on this date. If we do not hear from you within this time then it will be assumed that the paper has been withdrawn. In exceptional circumstances, extensions may be possible if agreed with the Editorial Office in advance. We do not allow multiple rounds of revision so we urge you to make every effort to fully address all of the comments at this stage. If deemed necessary by the Editors, your manuscript will be sent back to one or more of the original reviewers for assessment. If the original reviewers are not available we may invite new reviewers.

Please also include the following statements alongside the other end statements. As we cannot publish your manuscript without these end statements included, if you feel that a given heading is not relevant to your paper, please nevertheless include the heading and explicitly state that it is not relevant to your work.

- **Funding statement**

Please include a funding section after your main text which lists the source of funding for each author.

RSC Associate Editor:
Comments to the Author:
(There are no comments.)

RSC Subject Editor:
Comments to the Author:
(There are no comments.)

Reviewers' Comments to Author:
Reviewer: 1

Comments to the Author(s)

The manuscript reports carbon dots functionalized papers for high-throughput sensing of 4-Chloroethcathinone and its analogues in crime sites. There are some interesting results. And I think there is a wide reader region to have interests in this kind of material. I recommend accepting this work. I wish the authors could give some revision before it could be accepted.

- 1) Please explain the meaning of The Raman spectrum. The description of The Raman spectrum in the article is not detailed enough.
- 2) Characterization of CDs lacks FTIR and XPS data
- 3) The author referred the relationship of logarithm of $(F_0-F)/F_0$ of C-dots is linear with the concentration of 4-chloroethcathinone over the range from 2.5 to 15.0 Mm, but we can't get this relationship from the Fig.1A.
- 4) For "To support our suggestion, other.....they thus were reduced from a keto form to a hydroxyl form by C-dots.", the author should give the picture of the data "48%, 45%, 43%, 47%, and 34%", and the comparison of the published article to state the advantages.
- 5) For "To investigate an electron transfer mechanism occurring between 4-chloroethcathinone and the C-dots, the lifetimes of C-dots without with containing 4-chloroethcathine (15 mM) at emission excitation wavelength of 430/360 nm were determined to be 13 and 11 ns, respectively." ,the author should give the picture of the data "13 and 11 ns".
- 6) The author referred Based on the absorption, PL, and CV data, a sensing mechanism of C-dots for 4-chloroethcathinone is proposed as displayed in Scheme 1, but we can find the absorption data
- 7) The authors write "Moreover, to further support the electron transfer mechanism.....It is noted that sodium borohydride itself caused about 12% decrease in the PL intensity of C-dots." Please explain the mechanism of 4-chloroethcathinone reacted with sodium borohydride.

Reviewer: 2

Comments to the Author(s)

In this manuscript, Yen et al reported the preparation of a novel kind of C-dot functionalized paper that can be used for detecting 4-Chloroethcathinone and its analogues with a high

selectivity. The topic is interesting and the results sounds good. Unfortunately, I cannot accept this paper for publication at its present form, because the manuscript was poorly prepared, besides this, I also have some serious concerns about the data analysis as follows. In my opinion, a major revision should be done before it can be re-considered for publication in the Royal Society Open Science journal.

(1) On page 5, the authors claimed that “The TEM image of as-prepared C-dots displayed in Fig. S1 (Supporting Information) shows that they are uniform and monodispersed spheres with a mean diameter of 5 nm (100 counts)”. However, the authors did not provide the size distribution analysis curve on the basis of the TEM image shown in Fig. S1;

(2) Fig. S2 (Supporting Information) shows “the D-band at 1320 cm^{-1} and the G-band at 1590 cm^{-1} ”, however, there is an obvious Raman signal at 1450 cm^{-1} between D and G bands, what does this represent for? The authors need to give an assignment of this Raman signal.

(3) Still on page 5, “Their intensity ratio (ID/IG) is 1.4, which is similar to the most reported C-dots, supporting C-dots containing sp² (core) and sp³ (surface) hybridized carbons.[33-36]” I think the authors have missed many related papers in explanation of ID/IG ratio of carbon dots, because most of the carbon dots with sp²-hybridized carbon cores show the ID/IG ratios in the range of 0.3 ~ 0.8. And the listed references are not adequate here.

(4) On page 3, C-dots were synthesized from L-arginine aqueous solution in a Teflon-lined stainless-steel container through a hydrothermal route after heating at 240 °C for 14 h. It is reasonable to expect that the as-synthesized C-dots are highly crystallized at so high a temperature, however, The authors did not provide a high-resolution TEM image of the carbon core, and Raman spectrum of the C-dots as shown in Fig. S2 is very different from that of the purified carbon dots as previously reported. It suggests the presence of a sort of impurity inside the C-dots.

(5) On page 5, the authors thought D-band at 1320 cm^{-1} representing for the edge defects, but this is not correct. Since the edge defects usually exist in graphene quantum dots, herein no any evidence can demonstrate that the obtained C-dots are graphene quantum dots.

(6) Although using CV and UV-vis absorption methods to determine the HOMO and LUMO levels of the C-dots (Fig. 2A) is very interesting, according to Table 1, the band gap was evaluated to be 3.03 eV, this value should be compared to those obtained from theoretical calculation in the literature.

Author's Response to Decision Letter for (RSOS-191017.R0)

See Appendix A.

RSOS-191017.R1 (Revision)

Review form: Reviewer 1

Is the manuscript scientifically sound in its present form?

Yes

Are the interpretations and conclusions justified by the results?

Yes

Is the language acceptable?

Yes

Do you have any ethical concerns with this paper?

No

Have you any concerns about statistical analyses in this paper?

No

Recommendation?

Accept as is

Comments to the Author(s)

It can be accepted at present.

Review form: Reviewer 2

Is the manuscript scientifically sound in its present form?

Yes

Are the interpretations and conclusions justified by the results?

Yes

Is the language acceptable?

Yes

Do you have any ethical concerns with this paper?

No

Have you any concerns about statistical analyses in this paper?

No

Recommendation?

Accept as is

Comments to the Author(s)

The authors has reponded to all of the reviewers' comments point-by-point in their revisions. I recommend it to be accepted for publication at its present form.

Decision letter (RSOS-191017.R1)

12-Aug-2019

Dear Dr Chang:

Title: Carbon Dots Functionalized Papers for High-Throughput Sensing of 4-Chloroethcathinone and its analogues in Crime Sites

Manuscript ID: RSOS-191017.R1

It is a pleasure to accept your manuscript in its current form for publication in Royal Society Open Science. The chemistry content of Royal Society Open Science is published in collaboration with the Royal Society of Chemistry.

RSC Associate Editor:
Comments to the Author:
(There are no comments.)

RSC Subject Editor:
Comments to the Author:
(There are no comments.)

Reviewer(s)' Comments to Author:
Reviewer: 2

Comments to the Author(s)
The authors has reponded to all of the reviewers' comments point-by-point in their revisions. I recommend it to be accepted for publication at its present form.

Reviewer: 1

Comments to the Author(s)
It can be accepted at present.

Appendix A

Department of Chemistry
National Taiwan University
1, Section 4, Roosevelt Road
Taipei 106, Taiwan

July 19, 2019

Dr. Jeremy Sanders FRS
Chief Editor
Royal Society Open Science
Department of Chemistry, Lensfield Road, Cambridge
T: +44 (0) 1223 336300
Email: enquiries@ch.cam.ac.uk

Dear Prof. Jeremy Sanders FRS

We thank you and the reviewers for your valuable comments to our manuscript entitled “Carbon Dots Functionalized Papers for High-Throughput Sensing of 4-Chloroethcathinone and its analogues in Crime Sites (RSOS-191017). We have revised the manuscript according to the comments. The point-to-point changes are highlighted in yellow in the revised manuscript. We hope the revised manuscript can meet the high standard of *Royal Society Open Science*.

Sincerely yours,

Huan-Tsung Chang
Professor of Chemistry
National Taiwan University

changht@ntu.edu.tw

Responses to Reviewer's comments

Reviewer 1:

Specific comments:

1. Please explain the meaning of The Raman spectrum. The description of The Raman spectrum in the article is not detailed enough.

Response: A better quality of Raman spectrum of C-dots is provided in Fig. 1C in the revised manuscript. The signals of D-band and G-band at 1358 and 1572 cm^{-1} are assigned for the vibrations of carbon atoms with dangling bonds in the termination plane of disordered graphite or glassy carbon and the in-plane stretching of sp^2 carbon in the rings, respectively,

2. Characterization of CDs lacks FTIR and XPS data.

Response: We added FTIR and XPS data to Fig. S3 (supporting information), Fig. 1B and Fig. S2 (supporting information), respectively, as suggested.

3. The author referred the relationship of logarithm of $(F_0 - F)/F_0$ of C-dots is linear with the concentration of 4-chloroethcathinone over the range from 2.5 to 15.0 Mm, but we can't get this relationship from the Fig.1A.

Response: The sentence had been revised to "the relationship of $(F_0 - F)/F_0$ of C-dots is linear with the concentration of 4-chloroethcathinone over the range from 2.5 to 15.0 mM" as shown in the inset to Fig. 2A. The linear relationship is $Y = 0.028X + 0.023$ ($R^2 = 0.98$)

4. For "To support our suggestion, other.....they thus were reduced from a keto form to a hydroxyl form by C-dots." ,the author should give the picture of the data "48%, 45%, 43%, 47%, and 34%" , and the comparison of the published article to state

the advantages

Response: The quenching data for 4-chloromethcathinone, ethylone, butylone, mexedrone, acetone, formaldehyde, gamma-butyrolactone and 1-methyl-2-pyrrolidinone are added to Fig. 2D. Advantages of the present approach over the reported ones (Drug testing and analysis 2016, 8 (1), 136-140; Analytical chemistry 2014, 86 (19), 9985-9992.) include simplicity and low cost.

5. For “To investigate an electron transfer mechanism occurring between 4-chloroethcathinone and the C-dots, the lifetimes of C-dots without with containing 4-chloroethcathine (15 mM) at emission excitation wavelength of 430/360 nm were determined to be 13 and 11 ns, respectively.”, the author should give the picture of the data “13 and 11 ns”

Response: The lifetime decay curves of C-dots in the absence/presence of 4-chloroethcathine (15 mM) are provided in Fig. S6 (supporting information) as suggested.

6. The author referred Based on the absorption, PL, and CV data, a sensing mechanism of C-dots for 4-chloroethcathinone is proposed as displayed in Scheme 1, but we can find the absorption data

Response: The absorption spectrum of C-dots is displayed in the inset to Fig. 1D and that for cocaine, 4-chloroethcathinone, and ephedrine are shown in Fig. S7 (supporting information).

7. The authors write “Moreover, to further support the electron transfer mechanism.....It is noted that sodium borohydride itself caused about 12% decrease in the PL intensity of C-dots.” Please explain the mechanism of 4-

chloroethcathinone reacted with sodium borohydride.

Response: Sodium borohydride as a reducing reagent induced reduction of the keto groups of 4-chloroethcathinone and the surface residues like keto or aldehyde groups of C-dots through nucleophilic attack by the hydride anion. It is well known that many oxidized residues such as carboxylate, keto, aldehyde are existent on the surface of C-dots prepared through a hydrothermal route.

Reviewer 2

1. On page 5, the authors claimed that “The TEM image of as-prepared C-dots displayed in Fig. S1 (Supporting Information) shows that they are uniform and monodispersed spheres with a mean diameter of 5 nm (100 counts)”. However, the authors did not provide the size distribution analysis curve on the basis of the TEM image shown in Fig. S1.

Response: The size distribution of C-dots is added to Fig. 1A (inset) as suggested.

2. Fig. S2 (Supporting Information) shows “the D-band at 1320 cm^{-1} and the G-band at 1590 cm^{-1} ”, however, there is an obvious Raman signal at 1450 cm^{-1} between D and G bands, what does this represent for? The authors need to give an assignment of this Raman signal.

Response: A better quality of Raman spectrum of C-dots is provided in Fig. 1C in the revised manuscript. The signals of D-band and G-band at 1358 and 1572 cm^{-1} are assigned for the vibrations of carbon atoms with dangling bonds in the termination plane of disordered graphite or glassy carbon and the in-plane stretching of sp^2 carbon in the rings, respectively,

3. Still on page 5, “Their intensity ratio (I_D/I_G) is 1.4, which is similar to the most

reported C-dots, supporting C-dots containing sp² (core) and sp³ (surface) hybridized carbons.[33-36]” I think the authors have missed many related papers in explanation of I_D/I_G ratio of carbon dots, because most of the carbon dots with sp²-hybridized carbon cores show the I_D/I_G ratios in the range of 0.3 ~ 0.8. And the listed references are not adequate here.

Response: The I_D/I_G of C-dots is 0.8. More papers (New J. Chem. 38, 4946-4951; RSC Adv. 9, 8628-8637; Biosens. Bioelectron 74 (2015) 263–269; Adv. Mater. 2018, 30, 1704740) are cited.

4. On page 3, C-dots were synthesized from L-arginine aqueous solution in a Teflon-lined stainless-steel container through a hydrothermal route after heating at 240 °C for 14 h. It is reasonable to expect that the as-synthesized C-dots are highly crystallized at so high a temperature, however, The authors did not provide a high-resolution TEM image of the carbon core, and Raman spectrum of the C-dots as shown in Fig. S2 is very different from that of the purified carbon dots as previously reported. It suggests the presence of a sort of impurity inside the C-dots.

Response: As suggested, an HRTEM image of C-dots is added to Fig. S1 (supporting information) and a better quality of Raman spectrum is displayed in Fig. 1C. The fact that a lattice spacing of C-dots observed in the HRTEM supports the formation of highly crystallized C-dots.

5. On page 5, the authors thought D-band at 1320 cm⁻¹ representing for the edge defects, but this is not correct. Since the edge defects usually exist in graphene quantum dots, herein no any evidence can demonstrate that the obtained C-dots are graphene quantum dots.

Response: The D-band is mainly due to the vibrations of carbon atoms with

dangling bonds in the termination plane of disordered graphite or glassy carbon.

6. Although using CV and UV-vis absorption methods to determine the HOMO and LUMO levels of the C-dots (Fig. 2A) is very interesting, according to Table 1, the band gap was evaluated to be 3.03 eV, this value should be compared to those obtained from theoretical calculation in the literature.

Response: As suggested, the HOMO, LUMO, and energy gap of C-dots obtained from this study and that are acquired from a DFT calculation (Sci China Chem, 2018, 61, 491-495) are added in the revised manuscript. The values of HOMO (-4.51 eV), LUMO (-1.48 eV), and energy gap (3.03 eV) of C-dots acquired from our study are close to that (HOMO: -4.81 eV, LUMO: -1.74 eV, energy gap: 3.08 eV) reported in the literature.